# Oropharyngeal mucosal transmission of Zika virus in rhesus macaques

Christina M. Newman[1], Dawn M. Dudley[1], Matthew T. Aliota [2], Andrea M. Weiler[3], Gabrielle L. Barry[3], Mariel S. Mohns[1], Meghan E. Breitbach[1], Laurel M. Stewart[1], Connor R. Buechler[1], Michael E. Graham[1], Jennifer Post[3], Nancy Schultz-Darken[3], Eric Peterson[3], Wendy Newton[3], Emma L. Mohr[4], Saverio CapuanoIII[3], David H. O'Connor[1,3] & Thomas C. Friedrich[2,3]

Zika virus is present in urine, saliva, tears, and breast milk, but the transmission risk associated with these body fluids is currently unknown. Here we evaluate the risk of Zika virus transmission through mucosal contact in rhesus macaques. Application of high-dose Zika virus directly to the tonsils of three rhesus macaques results in detectable plasma viremia in all animals by 2 days post-exposure; virus replication kinetics are similar to those observed in animals infected subcutaneously. Three additional macaques inoculated subcutaneously with Zika virus served as saliva donors to assess the transmission risk from contact with oral secretions from an infected individual. Seven naive animals repeatedly exposed to donor saliva via the conjunctivae, tonsils, or nostrils did not become infected. Our results suggest that there is a risk of Zika virus transmission via the mucosal route, but that the risk posed by oral secretions from individuals with a typical course of Zika virus infection is low.

[1] Department of Pathology and Laboratory Medicine, University of Wisconsin School of Medicine and Public Health, 3170 UW Medical Foundation Centennial Building, 1685 Highland Ave., Madison, WI 53705, USA. [2] Department of Pathobiological Sciences, University of Wisconsin School of Veterinary Medicine, 2015 Linden Dr., Madison, WI 53706, USA. [3] Wisconsin National Primate Research Center, University of Wisconsin, 1220 Capitol Ct., Madison, WI 53715, USA. [4] Department of Pediatrics, University of Wisconsin, University of Wisconsin Clinical Science Center, 600 Highland Ave., Madison, WI 53792, USA. Christina M. Newman and Dawn M. Dudley contributed equally to this work. Correspondence and requests for materials should be addressed to T.C.F. (email: thomasf@primate.wisc.edu)

Zika virus (ZIKV) is a mosquito-borne flavivirus that is associated with Guillain–Barré syndrome in adults and a range of birth defects, most notably microcephaly, in congenitally infected infants[1–5]. ZIKV has become a subject of global concern as it has rapidly expanded its geographic range in the past 2 years. One of the many surprising aspects of the current ZIKV pandemic is the confirmation of one previous report of sexually transmitted ZIKV infection[6–9]; there is now evidence that ZIKV of both the Asian/American and African genetic lineages can be transmitted sexually. Because mosquito-borne flavivirus infection has not previously been associated with human-to-human transmission, understanding transmission risks is critical for designing effective prevention and control strategies.

In the Americas, ZIKV is transmitted among humans by *Aedes* species mosquitoes, primarily *Aedes aegypti*[10–13]. The explosive epidemic in tropical areas where important vector mosquito species are common has made it challenging to differentiate between vector-borne and sexual ZIKV transmission, so the risk of human-to-human transmission has been difficult to assess. In the continental United States, which has primarily travel-associated cases, detection of non-vector transmission is much more straightforward, and 46 cases of sexually transmitted ZIKV infection have been reported as of 20 April 2017 (https://www.cdc.gov/zika/geo/united-states.html). These cases have included male-to-female, male-to-male, and female-to-male transmission[6, 14–16].

ZIKV RNA has been detected in blood, semen, and vaginal secretions, consistent with observations of sexual transmission.

Viral RNA and/or infectious virus has also been reported in urine, saliva, tears, and breast milk, suggesting that these body fluids may also pose a transmission risk[17–20]. Indeed, virus was transmitted to a caregiver of an individual with high ZIKV viremia who eventually succumbed to infection[21].

Here we used rhesus macaques to evaluate the risk of ZIKV transmission through mucosal membrane contact with saliva (specifically, palatine tonsils, nasal mucosae, and conjunctivae). We applied high-dose ZIKV stock directly to the tonsils of one group of animals to assess what we term the "theoretical" risk of mucosal transmission using a dose ($8 \times 10^5$ PFU/ml) that is 20-fold higher than that typically found in saliva. In another experiment, we applied saliva from ZIKV-infected macaques to naive animals to determine whether ZIKV might also be transmitted by more casual contact. All animals that received the high dose of virus applied directly to the tonsils became infected with ZIKV. However, animals that were repeatedly challenged with saliva from ZIKV-positive animals remained uninfected. Together, these results suggest that, for a typical ZIKV infection, the risk of mucosal transmission is very low.

## Results

**ZIKV application to tonsils results in systemic infection.** To evaluate the risk of ZIKV transmission via the oropharyngeal mucosa, we applied $8 \times 10^5$ PFU Zika virus/H.sapiens-tc/FRA/2013/FrenchPolynesia-01-v1c1 (ZIKV-FP;[22]) directly to the palatine tonsils of three Indian-origin rhesus macaques using a

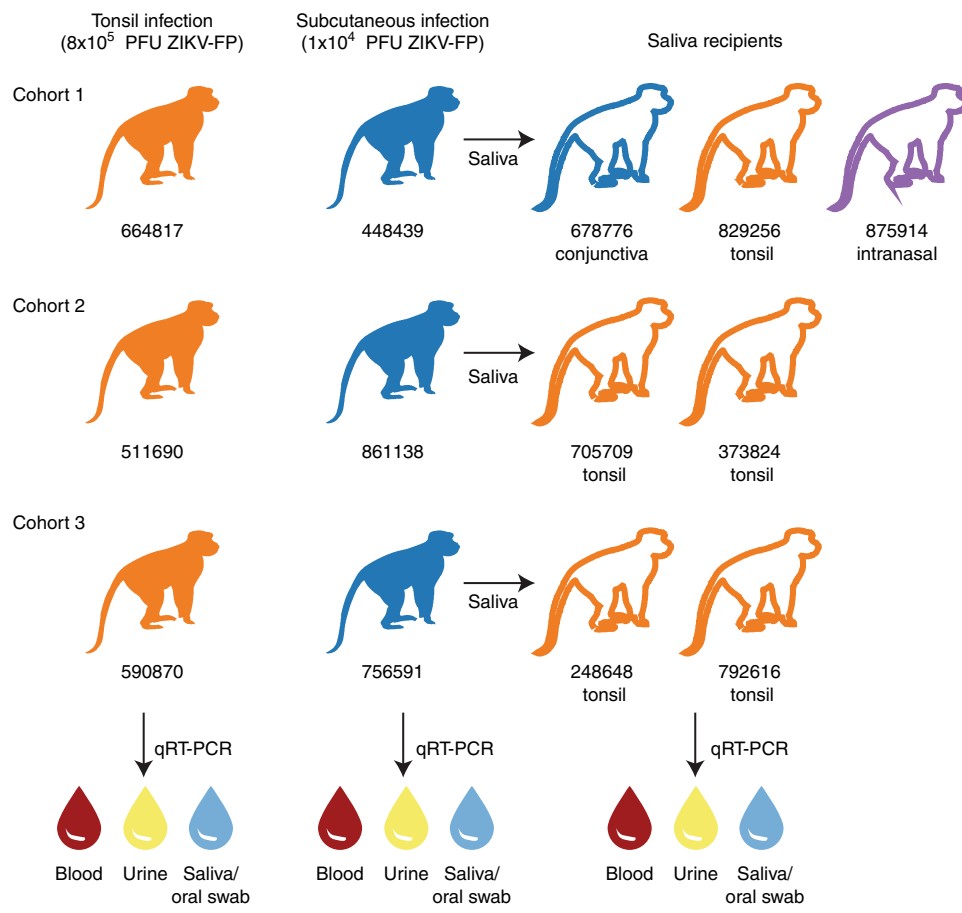

**Fig. 1** Study design. Two cohorts of animals were inoculated with Zika virus either by application of stock virus to the tonsils (*orange filled symbols*) or subcutaneously (*blue-filled symbols*). Saliva from animals infected subcutaneously was used to challenge naive recipient animals (*open symbols*) either to the palatine tonsils, conjunctivae or nasal passages. Blood plasma, urine and oral swabs (and/or saliva) were tested for Zika virus RNA by qRT-PCR in all animals

pipet (Fig. 1). One macaque infected by this route (664817) had detectable plasma viremia by 1 day post-infection (dpi) and all three had detectable plasma viremia by 2 days post-infection (dpi) (Fig. 2a, b). Peak plasma viremia was observed at 6 dpi in all three animals and ranged from $2.4 \times 10^5$ to $1.1 \times 10^7$ vRNA copies per ml. ZIKV plasma viremia was undetectable in all animals by 14 dpi. Plasma viremia of animals infected after tonsillar inoculation (Fig. 2b, orange traces) was compared to plasma

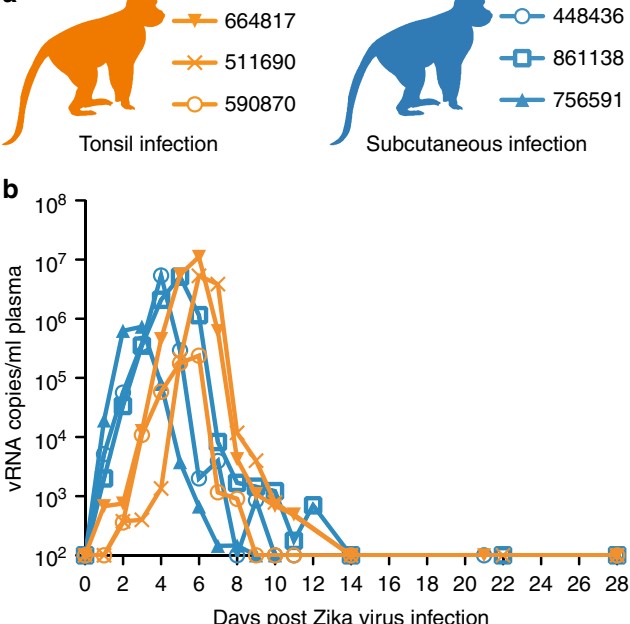

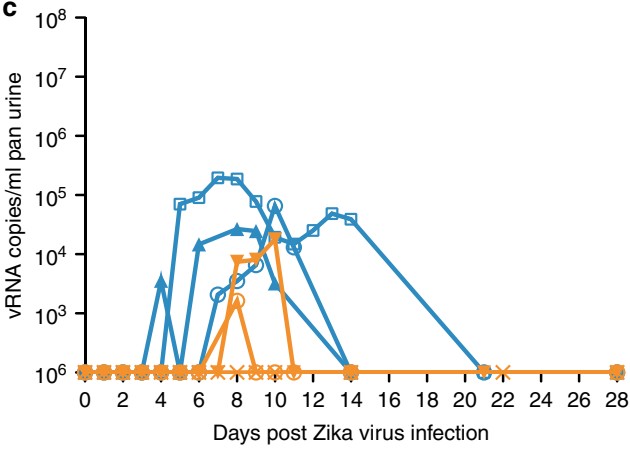

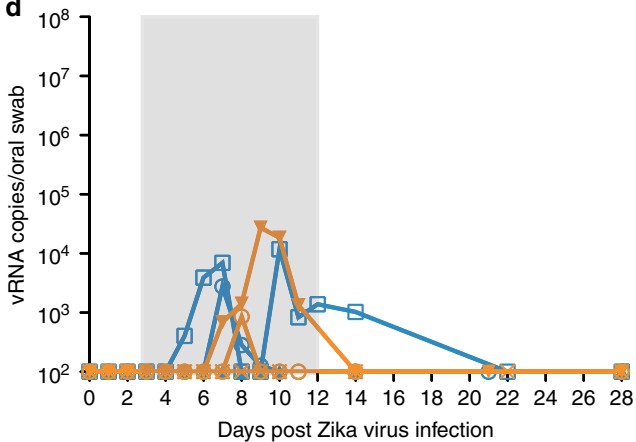

viremia of three animals infected subcutaneously with $1 \times 10^4$ PFU of ZIKV-FP (Fig. 2b, blue traces). These subcutaneously inoculated animals also served as saliva donors to recipient animals described below and in Fig. 1. Overall, plasma viremia in macaques infected via the oral mucosae was similar in magnitude and duration to that of macaques inoculated subcutaneously. Two notable exceptions were that viremia only became detectable 2 days after infection in two of the three macaques infected via the tonsils, and the peak plasma viral load in these animals occurred 1–3 days later than in animals infected subcutaneously (Fig. 2a, b).

ZIKV RNA was also detectable in other bodily fluids following tonsil exposure. ZIKV RNA was detected in the urine of animal 664817 at 8 dpi ($1.6 \times 10^3$ vRNA copies per ml) and 590870 at 8–10 dpi (range = $7.4 \times 10^3$–$1.8 \times 10^4$ vRNA copies per ml), but not in 511690 (Fig. 2a, c). We collected saliva when possible, but because the volume of saliva that could be collected was often insufficient to accurately assess viral load, we also collected oral secretions using absorbent swabs that were eluted in a standardized volume of viral transport media (VTM). ZIKV RNA detection in oral swab samples was variable from most animals and undetectable from animal 511690 (Fig. 2a, d). ZIKV RNA was detected in saliva from animal 590870 on days 6–9 post infection and from animal 511690 on days 8 and 10 post infection (Fig. 3). Overall, when saliva samples were available, ZIKV RNA was detected more consistently and at more time points than in oral swabs (Fig. 3). By 14 dpi, ZIKV RNA was undetectable in the tested body fluids of all animals infected by tonsil inoculation.

Sera from macaques that were infected by application of ZIKV to the tonsils neutralized ZIKV-FP across a range of serum dilutions. Indeed, neutralization curves prepared using sera from all 3 animals revealed a similar profile as compared to sera from animals infected subcutaneously (see Supplementary Fig. 1). All animals developed neutralizing antibodies (nAb) with a 90% plaque reduction neutralization test (PRNT$_{90}$) titer of 1:160 (664817 and 511690) or 1:80 (590870) by 28 dpi. Together, these results show that application of high doses of ZIKV to the oral mucosae can result in systemic infection and induce humoral immune responses in a manner similar to subcutaneous infection.

**Donor saliva does not induce systemic infection**. The highest concentration of ZIKV RNA we observed in the oral secretions of infected macaques in previous studies was $2.9 \times 10^4$ vRNA copies per ml, while the dose of ZIKV stock applied to the tonsils of the animals described above contained $8 \times 10^5$ PFU (~$8 \times 10^8$ vRNA copies per ml). To examine whether saliva from ZIKV-infected macaques was infectious and represented an actual transmission risk to uninfected macaques through oropharyngeal mucosal exposure, we collected saliva from the three subcutaneously inoculated rhesus macaques described above. These animals served as saliva donors to 7 naive recipients—saliva or oral swabs were collected from donors daily from 3–10 dpi (cohorts 1 and 3) or 3–12 dpi (cohort 2) and applied to the tonsils, nostrils, or conjunctivae of 1 or more recipients (See Fig. 1 for details). This

**Fig. 2** Longitudinal detection of ZIKV RNA in plasma, urine and oral swabs in subcutaneously infected animals (*blue*) or animals inoculated directly to the tonsils (*orange*). **a** Legend for graphs represented in parts **b**–**d**. *Orange* lines represent animals infected via the tonsils and *blue* lines represent animals infected via subcutaneous injection. **b** ZIKV RNA copies per ml of peripheral blood plasma. **c** ZIKV RNA copies per ml of passively collected urine from pans. **d** ZIKV RNA copies per oral swab. The *gray box* indicates the time frame in which saliva or an oral swab sample from saliva donor animals was used to challenge recipient animals. The *y* axis starts at the limit of quantification of the qRT-PCR assay (100 vRNA copies per ml)

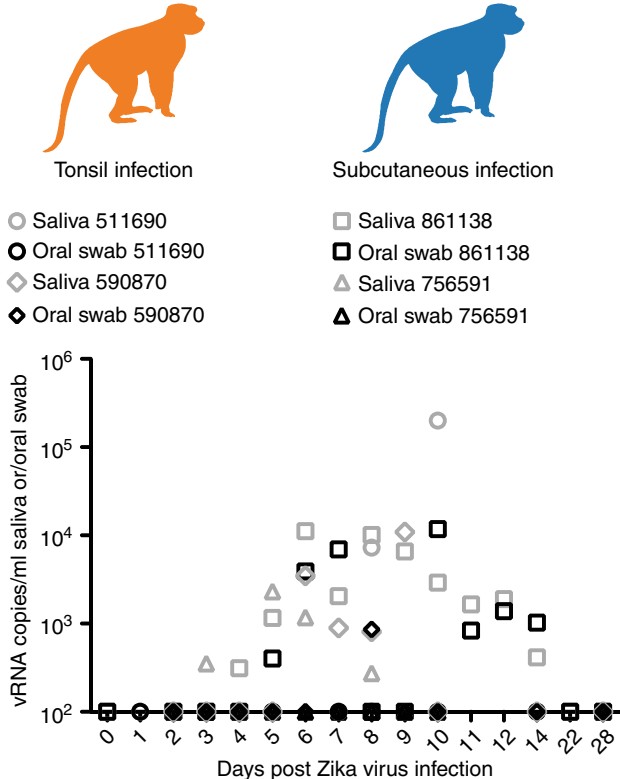

**Fig. 3** Longitudinal Zika virus load detected in saliva and oral swabs. Two of three animals infected either via the tonsils (under *orange monkey*) or subcutaneously (under *blue monkey*) had sufficient saliva collected for viral load testing. Only time points in these four animals at which both saliva (*gray*) and oral swabs (*black*) were tested simultaneously are shown. All points not distinguishable above the limit of quantification (100 vRNA copies per ml) are present on the *x* axis

3–10 or 3–12 dpi timeframe encompassed the period of time in which ZIKV RNA was detected in non-blood body fluids of animals infected subcutaneously in our previous studies[22, 23]. All saliva donors had detectable ZIKV plasma viremia by 1 dpi, peaking 3–5 dpi, (range = $7.35 \times 10^5$–$5.42 \times 10^6$ vRNA copies per ml) and resolving by 14 dpi (Fig. 2a, b). ZIKV RNA was detected in the passively collected urine of all 3 donors. Peak urine viral loads were detected 7–10 dpi (range = $2.68 \times 10^4$–$1.95 \times 10^5$ vRNA copies per ml) (Fig. 2c).

We monitored ZIKV RNA in saliva over time through collection of oral swabs and, whenever possible, saliva, from donor animals, to measure vRNA in the inoculae used to challenge the recipients (Fig. 2a, d, and 3). We detected ZIKV RNA in the oral swab eluate of two saliva donors (448436 and 861138); levels peaked at $2.77 \times 10^3$ vRNA copies per ml at 7 dpi and $1.18 \times 10^4$ vRNA copies per ml at 10 dpi, respectively. The last saliva donor (756591) did not have detectable ZIKV RNA in oral swab samples. We were unable to consistently collect saliva from 448436, but ZIKV RNA in the saliva of the 861138 was detectable 4–14 dpi and peaked at $1.12 \times 10^4$ vRNA copies per ml (Fig. 3). 756591 had detectable ZIKV RNA in saliva that peaked at $2.32 \times 10^3$ vRNA copies per ml at 5 dpi. Considered together, all donor animals had at least two timepoints with detectable vRNA in either the saliva or oral swab samples during the time when saliva was used to challenge the recipient animals. At 28 dpi, serum collected from all three donor animals yielded $PRNT_{90}$ values of 1:160, similar to the recipients of tonsillar challenges (see Supplementary Fig. 1 for neutralization curves). Importantly, ZIKV RNA was undetectable in all seven recipient animals in

all body fluids tested throughout the study. Furthermore, ZIKV-specific neutralizing antibodies were undetectable in recipient animals $\geq$21 days after the final challenge with saliva from infected donors (Supplementary Fig. 1).

**Infectivity of saliva from infected donors in IFNAR −/− mice**. Because no recipients of saliva from infected donors had detectable systemic ZIKV infections, we tested the infectivity of donor saliva in *IFNAR−/−* mice. These mice are highly susceptible to ZIKV infection[24, 25], even at challenge doses as low as $1 \times 10^2$ fluorescent focus units[24] which is approximately equivalent to $1 \times 10^2$ PFU (Michael Diamond, personal communication). We inoculated 5- to 6-week-old *IFNAR−/−* mice in the foot pad with ZIKV RNA-positive samples collected from two subcutaneously infected donor macaques. Saliva from 861138 collected at 10 dpi contained $1.1 \times 10^4$ vRNA copies per ml; because the ratio of ZIKV RNA content to PFU in our stock was approximately 1000:1[22, 23], we estimate that the 50 μl inoculum contained ~$1 \times 10^{-0.25}$ PFU. The second sample was oral swab eluate from 448436 collected at 7 dpi. This contained $2.8 \times 10^3$ vRNA copies per ml, so we estimate that the 50 μl inoculum administered to the foot pad contained ~$1 \times 10^{-0.86}$ PFU. We monitored mice for 25 days for weight loss and mortality. All control and experimental mice survived with no evidence of morbidity (Supplementary Fig. 2). Furthermore, ZIKV RNA was undetectable in mouse serum at 3 dpi (the approximate peak of viremia in mice infected previously[24, 25]) and additional serum collected from these animals at 25 dpi did not inhibit in vitro replication of ZIKV, suggesting that they did not seroconvert. These data suggest that saliva from ZIKV-infected macaques did not contain enough replication-competent virus to initiate infection in these sensitive hosts. The same saliva and oral swab samples were concurrently inoculated onto Vero cells to attempt to quantify the infectious dose the mice might have received; however, both the saliva and oral swab sample used for the mouse challenge experiments gave no plaques on these cells (Supplementary Table 1). Together, these results suggest that macaques with ZIKV virus loads in the usual range after subcutaneous infection shed very small quantities of infectious ZIKV in saliva.

**Saliva reduces the infectivity of ZIKV in Vero cells**. To evaluate whether antiviral components of saliva reduce the infectivity of ZIKV, we performed a dose titration experiment using freshly collected saliva from flavivirus-naive macaques. ZIKV-FP was mixed with saliva at doses ranging from $1 \times 10^2$ to $8 \times 10^5$ PFU and assessed for infectivity by plaque assay on Vero cells. Virus diluted in PBS at the same concentrations was used as positive controls for comparison. The titers for virus diluted in freshly collected macaque saliva were 1–2 logs lower than for virus diluted in a comparable volume of PBS (Supplementary Fig. 3). A linear regression model was used to examine the difference in titer by plaque assay between ZIKV diluted in saliva and ZIKV diluted in PBS. On the basis of this regression model, diluting ZIKV in saliva results in a statistically significant reduction in the number of PFU at all dilutions compared to PBS (coefficient estimate = −2.6, SE = 0.4, t = −6.1, $P < 0.001$). Overall, the treatment group (saliva or PBS) and sample dilution explained 97% of the observed variation in log PFU (based on an $R^2$ value for the model of 0.97). These results suggest that one or more factors in macaque saliva may reduce ZIKV infectivity.

**Discussion**

Humans and animals infected with ZIKV are known to shed virus (or vRNA) in multiple body fluids, including blood, urine, saliva, and genital secretions[22, 26–28]. Sexual transmission of ZIKV

between humans has been documented in several cases, but the risk of transmission from more casual contact has been difficult to evaluate. Human-to-human transmission via non-sexual contact has been reported in a single case to date. This case involved a source patient with extremely high viremia; contact with tears or sweat from the source patient was hypothesized to be the mode of transmission[21]. Here we used a nonhuman primate model to investigate whether exposure to ZIKV via mucous membranes, particularly the oropharyngeal mucosae, represents an infection risk.

Application of a high dose ($8 \times 10^5$ PFU) of Asian-lineage ZIKV to the palatine tonsils resulted in systemic infection in three of three rhesus macaques, suggesting that productive infection can indeed be initiated at the oropharyngeal mucosae. The kinetics and magnitude of ZIKV replication in plasma in the threemucosally-infected animals were similar to those observed in animals infected subcutaneously with the same stock of ZIKV-FP (Fig. 2a, b), and all three animals developed strong nAb titers against homologous ZIKV-FP by 28 dpi. Detection of ZIKV RNA in oral swabs, saliva, and urine was variable in the animals that received a direct tonsil inoculation, findings that were also similar to those from animals infected subcutaneously in this and a previous study[22].

Limitations associated with sample collection could have influenced our ability to detect vRNA in fluids other than blood. For example, time from urination to sample collection varied because urine was collected passively from pans in the animals' housing. Similarly, there is wide variation in the degree to which individual animals salivate while under sedation. As a result, it was not always possible to collect saliva from each animal at each time point. Oral swabs provided a more consistent means for collecting oral secretions, but as the relatively small amount of secretions absorbed by the swabs must be eluted in medium, detection of ZIKV RNA or infectivity may not be as sensitive in swab sample as in undiluted saliva. Accordingly, ZIKV RNA was more consistently detected in the saliva vs. oral swabs when both sample types were available (Fig. 3). Although ZIKV RNA was detectable in the saliva of ZIKV donor animals 861138 and 756591 and in oral swab samples from 861138 and 448436, transfer of saliva from donors to the mucosae of naive recipients did not result in ZIKV transmission. In addition, donor saliva produced no detectable plaques on Vero cells. Moreover, IFNAR−/− mice inoculated with saliva or oral swab samples from infected donor macaques showed no overt signs of disease and did not seroconvert. Our previous studies suggest that the ratio of vRNA copies to infectious particles from both virus stock and sera from infected animals is ~1000:1[22, 23]. The highest viral load that we detected in a saliva sample in this study was only $1.1 \times 10^4$ vRNA copies per ml; given that we were never able to collect a full 1 ml of saliva, it seems likely that saliva transferred to recipients (tonsil maximum of 100–200 µl, conjunctivae maximum of 50 µl, and nasal passage maximum of 100 µl) in our study likely contained less than three infectious ZIKV virions.

In addition to sample collection limitations, saliva may represent a natural barrier to virus transmission in the oral cavity. A number of different viruses, such as HIV and influenza A, are detected in the saliva of infected individuals, but oral mucosal infection in many cases is considered low risk[29–31]. This may be due in part to the antimicrobial constituents present in saliva, which include mucins, lysozyme, peroxidase, defensins, and salivary agglutinin[31]. However, even with these natural barriers to infection, challenge of animals with a higher dose of ZIKV directly to the tonsils resulted in productive infection. Investigators have reported that a bolus challenge of SIV directly to the oral cavity can infect rhesus macaques, which they considered might approximate HIV transmission through oral-genital

contact[32]. The finding that an oral bolus of virus may overcome natural mucosal barriers to infection is especially interesting because ZIKV has been detected in semen for many months following acute infection and at viral loads significantly higher than those detected in blood plasma (8.6 $\log_{10}$ copies per ml)[33, 34]. ZIKV RNA and infectious virus have also been detected in breast milk, another potential route of oral mucosal exposure[35]. The potential for transmission of HIV through breast milk is well documented in humans and is associated with the milk viral load[36, 37]. Although there have been no documented cases of transmission between a nursing mother and her infant, a ZIKV viral load of greater than $8 \times 10^6$ vRNA copies per ml has been reported in the breast milk in a woman with acute ZIKV infection[18, 20].

Taken together, our results suggest that there is a risk of ZIKV transmission via the oral mucosal route, as shown by systemic infection in three of three macaques after application of high-dose infectious virus to the tonsils. However, the actual risk of transmission via mucosal exposure to saliva may be low—saliva from donor animals with typical plasma viremia harbored little or no infectious virus and could not mediate transmission of ZIKV in our study. However, it must be noted that secretions, including saliva from individuals with unusually high viral load, semen, and breast milk, could pose a transmission risk.

## Methods

**Study design.** This study was designed as a proof of concept study to examine whether ZIKV transmission may occur in the absence of vector-borne or sexual transmission, via saliva, in the rhesus macaque model. Nothing is currently known about the potential saliva transmission of ZIKV in vivo so we selected 3 animals for direct application of ZIKV stock to the palatine tonsils, three animals for subcutaneous inoculation with a well characterized dose of ZIKV[21] to serve as saliva donors, and 7 animals (a minimum of two per donor) to serve as recipients of saliva from infected donors (Fig. 1). Available animals were allocated to experimental groups based on qualitative assessment of salivation while under ketamine sedation as communicated by staff at the Wisconsin National Primate Research Center. Investigators were not blinded to experimental groups.

**Ethical approval.** This study was approved by the University of Wisconsin-Madison Institutional Animal Care and Use Committee (Animal Care and Use Protocol Number G005401 and V5519).

**Nonhuman primates.** Six male and seven female Indian-origin rhesus macaques (*Macaca mulatta*) utilized in this study were cared for by the staff at the Wisconsin National Primate Research Center (WNPRC) in accordance with the regulations, guidelines, and recommendations outlined in the Animal Welfare Act, the Guide for the Care and Use of Laboratory Animals, and the Weatherall report. In addition, all macaques utilized in the study were free of Macacineherpesvirus 1, simian retrovirus type D, simian T-lymphotropic virus type 1, simian immunodeficiency virus, and had no history of exposure to any dengue virus serotype. Animals ranged in age from 3 years to 17 years old (mean = 7.8 years). For all procedures, animals were anesthetized with an intramuscular dose of ketamine (10 ml/kg). Blood samples were obtained using a vacutainer or needle and syringe from the femoral or saphenous vein.

**Virus.** Macaques in this study were inoculated with Asian-lineage Zika virus/H. sapiens-tc/FRA/2013/FrenchPolynesia-01_v1c1 (ZIKV-FP) obtained from Xavier de Lamballerie (European Virus Archive, Marseille, France). This virus was originally isolated from a 51-year-old female in France after travel to French Polynesia in 2013 and passaged a single time on Vero cells (African green monkey kidney cells; CCL-81). This Asian lineage ZIKV isolate shares a recent common ancestor with strains currently circulating in the Americas and shares 99% nucleotide identity with them[38]. In addition, all ZIKV strains associated with epidemic urban, mosquito-borne transmission, sexual transmission, Guillain–Barré syndrome, and congenital Zika syndrome include a S139N in the membrane (M) protein, D683E in the envelope(E) protein, and V763M and T777M in the transmembrane domain of the E protein[39]. These changes are shared between the French Polynesian strain used in this study and the virus isolated from a fatal case of ZIKV described in Utah[21]. The Utah ZIKV isolate comes from the first documented non-vector, non-sexual, non-vertical transmission of ZIKV and shares >99.5% similarity at the nucleotide level with ZIKV-FP. The virus stock used in this study was prepared by inoculation onto a confluent monolayer of C6/36 mosquito cells (*Aedes albopictus* larval cells; CRL-1660). Cell lines were obtained from American Type Culture

Collection (ATCC), were not further authenticated, and were not specifically tested for mycoplasma. A single, clarified harvest of virus, with a titer of $5.9 \times 10^6$ PFU/ml ($3.9 \times 10^9$ vRNA copies per ml) ZIKV-FP was used as stock for all subcutaneous inoculations.

**Tonsil challenges.** The ZIKV-FP stock was thawed, diluted in PBS to $1 \times 10^6$ PFU/ml, and maintained on ice until inoculation. Each animal was anesthetized and $8 \times 10^5$ PFU ZIKV-FP was applied directly to the palatine tonsils (maximum of 400 μl per tonsil) via pipet after visualization with a laryngoscope. Animals were closely monitored by veterinary and animal care staff for adverse reactions and signs of disease. Animals were examined, and blood, urine, oral swabs, and saliva were collected from each animal daily from 1 through up to 12 dpi and then weekly thereafter through 28 dpi.

**Subcutaneous inoculations.** The ZIKV-FP stock was thawed, diluted in PBS to $1 \times 10^4$ PFU/ml, and loaded into a 3 ml syringe maintained on ice until inoculation. For subcutaneous inoculations, each of three Indian-origin rhesus macaques was anesthetized and inoculated subcutaneously over the cranial dorsum with 1 mL virus stock containing $1 \times 10^4$ PFU. All animals were closely monitored by veterinary and animal care staff for adverse reactions and signs of disease. Animals were examined, and blood, urine, oral swabs, and saliva were collected from each animal daily from 1 through up to 12 dpi and then weekly thereafter through 28 dpi.

**Mucosal membrane challenges.** Oral swabs or saliva were collected from subcutaneously inoculated, ZIKV-infected donor macaques for mucosal membrane challenge of uninfected recipient animals beginning at day three post-infection and continuing through day 10 (cohorts 1 and 3) or 12 (cohort 2) post-infection of the donor animals. For recipients challenged with saliva from the first donor, oral swabs were held under the tongue of the anesthetized donor animal for up to 2 min to collect saliva. Swabs were then placed in 750 μl of viral transport media (tissue culture medium 199 supplemented with 0.5% FBS and 1% antibiotic/antimycotic), swished vigorously for at least 15 s, and then swabs were discarded. Viral transport media containing saliva was then applied via pipet to the palatine tonsils (100 μl per tonsil) (after visualization via laryngoscope), conjunctivae (50 μl per eye), or nasal passages (100 μl per nostril) of the corresponding recipient animal. For recipients challenged with saliva from the second and third donors, saliva was collected via pipet from under the tongue of an anesthetized donor animal and applied directly via pipet to the palatine tonsils (200 μl per tonsil) of each recipient animal. Mucosal challenges were repeated daily for up to 10 days. Animals were closely monitored by veterinary and animal care staff for adverse reactions and signs of disease. As described previously, animals were examined, and blood, urine, saliva, and oral swabs were collected from each animal daily during challenge and then weekly thereafter through 28 days after final challenge.

**Plaque reduction neutralization test (PRNT).** Macaque serum samples were screened for ZIKV neutralizing antibody utilizing a plaque reduction neutralization test (PRNT) on Vero cells (ATCC #CCL-81). Endpoint titrations of reactive sera, using a 90% cut off (PRNT$_{90}$), were performed as described in ref. [40] against ZIKV-FP. Neutralization curves were generated using GraphPad Prism software. The resulting data were analyzed by non-linear regression to estimate the dilution of serum required to inhibit 50% of infection.

**Live virus isolation.** Mice deficient in the alpha/beta interferon receptor on the C57BL/6 background (*IFNAR−/−*) originated from embryos that were provided as a generous gift from Dr. Eva Harris (University of California, Berkley, CA) and were bred in the pathogen-free animal facilities of the University of Wisconsin-Madison School of Veterinary Medicine (Madison, WI). Four groups ($n = 12$) of 5- to 6-week-old, mixed sex mice were used to test infectivity of ZIKV RNA in macaque saliva. Mice were inoculated in the left, hind foot pad with 50 μl of saliva or oral swab eluate in viral transport media (VTM) from samples that tested positive for ZIKV RNA by qRT-PCR. Mock-infected experimental control mice received saliva or saliva swab eluate in VTM collected from ZIKV-negative control macaques. Positive control mice received $10^4$ PFU of the same ZIKV-FP stock that was used to infect the macaques. Following inoculation, mice were monitored twice daily for the duration of the study. Sub-mandibular blood draws were performed and serum was collected to verify viremia via qRT-PCR and nAb titers via PRNT. Concurrent with mouse infectivity assays, the remainder of the ZIKV RNA-positive saliva and oral swab samples were also assessed for infectivity by plaque assay on Vero cells. Duplicate wells were infected with 0.1 ml aliquots from serial 10-fold dilutions of oral swab eluate or saliva in growth media and allowed to adsorb for 1 h. Following incubation, the inoculum was removed, and monolayers were overlaid with 3 ml containing a 1:1 mixture of 1.2% oxoid agar and $2 \times$ DMEM (Gibco, Carlsbad, CA, USA) with 10% (vol/vol) FBS and 2% (vol/vol) penicillin/ streptomycin. Cells were incubated at 37 °C in 5% CO$_2$ for 4 days for plaque development. Cell monolayers then were stained with 3 ml of overlay containing a 1:1 mixture of 1.2% oxoid agar and $2 \times$ DMEM with 2% (vol/vol) FBS, 2% (vol/vol) penicillin/streptomycin and 0.33% neutral red (Gibco). Cells were incubated overnight at 37 °C and plaques were counted.

**In vitro saliva titration.** Fresh saliva collected from uninfected macaques was tested for its capacity to inhibit the infectivity of ZIKV-FP in Vero cells. ZIKV was mixed with saliva 1:10 or 1:3 for 1 h at 37 C prior to being added to Vero cells. Final concentrations were $1 \times 10^2$–$1 \times 10^5$ PFU for the 1:10 dilutions and $8 \times 10^5$ PFU for the 1:3 dilution. Infection was measured by plaque assay as described previously. Phosphate buffered saline (PBS) spiked with the same concentrations of ZIKV-FP were used as positive controls. Results from two biological replicates (containing 5 dilutions each) were combined because, based on Student's *t*-tests, the variation within each treatment group was not significantly different between replicates ($t = 0.41$, df = 6.5, $P = 0.70$ for PBS and $t = −0.053$, df = 7.9, $P = 0.96$ for saliva). Plaque forming units were log transformed to meet the assumption of normality with respect to the explanatory variables prior to analysis. Based on an F-test, treatment group variance did not differ significantly between groups ($F = 1.3$, df = 4, 4, $P = 0.81$). Treatment groups were compared using a linear regression model with log PFU as the response variable and treatment group (saliva or PBS), and dilution as the explanatory variables in R version 3.3.2[41].

**Viral RNA isolation.** Plasma was isolated from EDTA-anticoagulated whole blood collected the same day by Ficoll density centrifugation at $1860 \times g$ for 30 min. Plasma was removed to a clean 15 ml conical tube and centrifuged at $670 \times g$ for an additional 8 min to remove residual cells. Urine was opportunistically collected from a pan beneath each animal's cage and centrifuged at $500 \times g$ for 5 min to remove cells and debris. Saliva was collected via pipet from under the tongue or using sterile oral swabs run under the tongue while animals were anesthetized. Saliva collected via pipet was used to directly inoculate recipient animals in cohorts 2 and 3. Additional saliva, when available, was removed from the collection pipet, centrifuged at $500 \times g$ for 5 min, and added to 650 μl viral transport media at a ratio of no more than one part saliva to three parts viral transport media. Swabs were placed in viral transport media for 60–90 min, then vortexed vigorously and centrifuged at $500 \times g$ for 5 min to elute saliva. Prior to extraction, oral swab eluates were pelleted by centrifugation at 14,000 rpm and 4 °C for an hour. After centrifugation, supernatant was removed, leaving virus in 200 μl media. Viral RNA was extracted from 300 μl plasma or urine using the Viral Total Nucleic Acid Kit (Promega, Madison, WI) on a Maxwell 16 MDx instrument (Promega, Madison, WI). Viral RNA was extracted from 200 μl oral swab-derived samples using the QIAamp MinElute Virus Spin Kit (Qiagen, Germantown, MD) with all optional washes. RNA was then quantified using quantitative RT-PCR. The viral load data from plasma and urine are expressed as vRNA copies/ml. The viral load data from oral swabs are expressed as vRNA copies/ml eluate. The viral load data from saliva in viral transport media are expressed as vRNA copies per ml total volume.

**Quantitative reverse transcription PCR (qRT-PCR).** For ZIKV-FP, vRNA from plasma, urine, saliva, and oral swabs was quantified by qRT-PCR using primers with a slight modification to those described by Lanciotti et al. to accommodate African lineage ZIKV sequences[42]. The modified primer sequences are: forward 5′-CGYTGCCCAACACAAGG-3′, reverse 5′-CACYAAYGTTCTTTTGCABA-CAT-3′, and probe 5′-6fam-AGCCTACCTTGAYAAGCARTCAGACACYCAA-BHQ1-3′. The RT-PCR was performed using the SuperScript III Platinum One-Step Quantitative RT-PCR system (Invitrogen, Carlsbad, CA) on a Light-Cycler 480 instrument (Roche Diagnostics, Indianapolis, IN). The primers and probe were used at final concentrations of 600 and 100 nm, respectively, along with 150 ng random primers (Promega, Madison, WI). Cycling conditions were as follows: 37 °C for 15 min, 50 °C for 30 min and 95 °C for 2 min, followed by 50 cycles of 95 °C for 15 s and 60 °C for 1 min. Viral RNA concentration was determined by interpolation onto an internal standard curve composed of seven 10-fold serial dilutions of a synthetic ZIKV RNA fragment based on ZIKV-FP.

**Data availability.** The primary data that support the findings of this study are available on the Zika Open-Research Portal (https://zika.labkey.com). The authors declare that all other data supporting the findings of this study are available within the article and its Supplementary Information files, or from the corresponding author upon request.

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

## Acknowledgements

We thank the Veterinary, Animal Care and Scientific Protocol Implementation staff at the Wisconsin National Primate Research Center (WNPRC) for their contribution to this study. We thank Emma Walker for assistance with the mouse studies. We acknowledge Jens Kuhn (NIAID/IRF, Frederick, MD) and Jiro Wada (US HHS/NIH, Washington, D. C., MD.) for preparing silhouettes of macaques used in figures. Funding for this project came from DHHS/PHS/NIH R01AI116382 to D.H.O. and from P51OD011106 awarded to the WNPRC, Madison-Wisconsin. This research was conducted in part at a facility constructed with support from Research Facilities Improvement Program grants RR15459-01 and RR020141-01. The publication's contents are solely the responsibility of the authors and do not necessarily represent the official views of NCRR or NIH.

## Author contributions

T.C.F., C.M.N., D.M.D., M.T.A. and D.H.O. designed the experiments. C.M.N., D.M.D., M.T.A. and T.C.F. analyzed the data and drafted the manuscript. M.T.A. provided and prepared viral stocks, performed plaque assays, and performed the mouse experiments. A.M.W., G.L.B. and T.C.F. developed and performed viral load assays. M.S.M., M.E.B., L.M.S., C.R.B, C.M.N., E.L.M. and D.M.D. coordinated and processed macaque samples for distribution. M.E.G. maintained the Zika Open Portal site where data were stored and shared. J.P., N.S.-D., E.P., S.C. and W.N., coordinated and performed macaque infections and sampling.

## Additional information

**Competing interests:** The authors declare no competing financial interests.

