## [Peer Review File · Nature Communications]

Reviewers' comments:

Reviewer #1 (Remarks to the Author):

The manuscript by Newman et al reports proof of concept that Zika may be acquired via the oropharyngeal route but does not provide a clear understanding of the likelihood of such transmission, since the success of transmission required relatively high virus input. While the demonstration is not to be minimized, the dose of virus corresponding to $\sim 4 \times 10^9$ viral RNA copies far exceeds any viral loads reported for any body fluids in humans or monkeys. The ideal experiment would have been to titrate the inoculum to provide a more realistic input into the probability of transmission.

The authors, also tested whether virus could be transmitted from the saliva. While the effort failed, these studies provided interesting insights from the fact that infectious Zika could not be found from this body fluid. I would strongly suggest the authors to spike normal macaque saliva with a dose range of their stock virus to test whether indeed salivary components are able to inhibit Zika or not. While this may not provide definite proof, it might point to potential mechanisms. Moreover, comparing infectious titers relative to vRNA copies from various body fluids in their model would provide a better evaluation of the model.

Otherwise the manuscript is well written and I had no additional points.

Reviewer #2 (Remarks to the Author):

Not only has Zika virus been shown to be transmitted through the bite of infected mosquitoes but also through sexual and casual contact with various bodily fluids (saliva, tears, semen etc.). However, the ZIKV transmission risk associated with casual and sexual contact is still unknown. In the current manuscript Newman et al. describe their findings in establishing an NHP model of oropharyngeal mucosal transmission of ZIKV. They demonstrate that applying ZIKV directly on the tonsil regions established a systemic infection in all three of their rhesus macaques. The kinetics of infection and antiviral antibody development were similar to NHP infected by the subcutaneous route. This finding indicates that the oral route of infection is possible. However, to determine whether virus can be transmitted in saliva from one animal to another, they transferred saliva from the SubQ infected animals, at times when saliva viral loads were highest, to naïve animals via oral administration. None of the animals challenged in this manner became infected. There are a number of potential reasons why virus failed to take hold via this mode of transmission. The authors challenged highly sensitive mice with saliva from the Zika virus infected animals and they also failed to succumb to infection. Thus it would appear that NHP can be infected via oropharyngeal route but at least in NHP, the low infectious viral load in saliva would preclude this from actually happening. The study is important, provocative and timely. The information clearly shows that oral infection is possible, however, a few concerns should be considered.

1. Previously the authors performed a very informative study wherein they titrated the

amount of inoculum used to subcutaneously infect RM. This was a critical study that demonstrated for the first time the range of virus amount necessary for transmission. Because the current study is similar in nature, as they are developing a novel model of ZIKV transmission, it would have been very helpful to the research community if they had determined the minimal amount of infectious viral particles it takes to transmit via the oral route. This information would critically guide their studies on saliva transfer. For example, if it takes at least 1,000 infectious units to infect through the oral route and you can't get that much virus in the saliva, maybe this would have guided these experiments and/or at least defined why transmission failed to happen.

2. Related to this, are there components in saliva that prohibit viral transmission or inactivate ZIKV. In vitro or mouse experiments utilizing RM saliva spiked with known titers of virus would shed light on this and help to explain why they failed to see transfer of virus to mice or Vero cells. Maybe the results could be compared to urine as another accessible bodily fluid for this type of study.

3. The authors should justify the virus strain that was used. Was there an association of this virus with casual or sexual contact?

4. The layout of the paper results is a little hard to follow. Might it be better to first directly compare the results from the two challenge models (subQ vs. oral)? And then talk about the saliva inoculations. As is, the results from the SubQ come between the Oral and saliva challenge. At first glance the reader is inclined to think that an additional SubQ cohort was used for these studies but this is not the case.

5. The authors demonstrate nicely that the neutralizing titers at 28 dpi are similar between the SubQ and oral infection routes. However, the viremia seemed to lag a bit at early times post infection. Did the antibody responses also lag with similar kinetics? Analysis of antibody responses at additional early time points, by ELISA or neutralization assays, would help to determine whether the response times for Ab development were the same or offset by the viral lag period. In addition pre-infection serum for each animal should be used as a control.

6. Were the first two groups of animals taken to necropsy to determine sites of virus infection? If so this data should be incorporated into the manuscript.

7. Often times in the text Figure references are not accurate. For example, lines 70, 79, 116—Fig. 2a should be Fig. 2b. There are similar discrepancies with other figure references within the text.

8. While it is negative, the Ab titers data for the saliva challenged animals should be shown to prove to the reader that they did not become infected.

Reviewer #1 (Remarks to the Author):

The manuscript by Newman et al reports proof of concept that Zika may be acquired via the oropharyngeal route but does not provide a clear understanding of the likelihood of such transmission, since the success of transmission required relatively high virus input. While the demonstration is not to be minimized, the dose of virus corresponding to $\sim 4 \times 10^9$ viral RNA copies far exceeds any viral loads reported for any body fluids in humans or monkeys. The ideal experiment would have been to titrate the inoculum to provide a more realistic input into the probability of transmission.

The reviewer makes a good point that we were not able to determine the minimal dose sufficient for infection by the oropharyngeal route. The high dose was chosen to model the unusual human case report of non-sexual, non-mosquito-borne transmission, which was associated with extraordinarily high viral loads in the index case. This establishes that very-high-titer ZIKV replication can occur, albeit rarely, in infected humans. Nonetheless, the reviewer is correct that our study represents a limitation of the nonhuman primate model: many more animals would be required to titrate the dose and infect at least 3 animals per dose for reproducibility and minimal statistical significance. Rather than titrate, we chose to test completely physiological conditions by determining whether saliva from infected animals could transmit ZIKV to naive recipients. This was the most direct way to determine the risk of acquisition via casual contact in the most physiologically relevant model while minimizing the number of animals used.

The authors, also tested whether virus could be transmitted from the saliva. While the effort failed, these studies provided interesting insights from the fact that infectious Zika could not be found from this body fluid. I would strongly suggest the authors to spike normal macaque saliva with a dose range of their stock virus to test whether indeed salivary components are able to inhibit Zika or not. While this may not provide definite proof, it might point to potential mechanisms. Moreover, comparing infectious titers relative to vRNA copies from various body fluids in their model would provide a better evaluation of the model.

This is a great suggestion. We have performed this experiment and have presented it in the revised manuscript as Supplementary Figure 3. Plaque assays show that the infectious virus titer is more than 1 log lower when spiked into saliva than when the same dose of virus is diluted in PBS. This indicates that some factor or factors in saliva may inhibit virus replication and further explains why replicating virus could not be found in the saliva of our animals, especially given the low viral vRNA copy numbers and limited sample sizes.

Reviewer #2 (Remarks to the Author):

The study is important, provocative and timely. The information clearly shows that oral infection is possible, however, a few concerns should be considered.

1. Previously the authors performed a very informative study wherein they titered the amount of inoculum used to subcutaneously infect RM. This was a critical study that demonstrated for the first time the range of virus amount necessary for transmission. Because the current study is similar in nature, as they are developing a novel model of ZIKV transmission, it would have been very helpful to the research community if they had determined the minimal amount of infectious viral particles it takes to transmit via the oral route. This information would critically guide their studies on saliva transfer. For example, if it takes at least 1,000 infectious units to infect through the oral route and you can't get that much virus in the saliva, maybe this would have guided these experiments and/or at least defined why transmission failed to happen.

We understand this reviewer's concern, which was also raised by reviewer 1. Again, we were very limited in the animal numbers we could use for this project and chose to prioritize testing physiological conditions over titrations. Unfortunately, we cannot perform the requested titrations in nonhuman primates with current funding and animal limitations.

2. Related to this, are there components in saliva that prohibit viral transmission or inactivate ZIKV. In vitro or mouse experiments utilizing RM saliva spiked with known titers of virus would shed light on this and help to explain why they failed to see transfer of virus to mice or Vero cells. Maybe the results could be compared to urine as another accessible bodily fluid for this type of study.

As addressed for reviewer 1, we have now performed an in vitro titration study by spiking virus into saliva from a ZIKV-naive animal. There does appear to be additional inhibition of virus in saliva relative to PBS. This data has now been added to the manuscript as Supplementary Figure 3. We do not think that comparing the virus replication in saliva to that in urine is a useful comparison because each fluid is very different in both viscosity and inhibitory components.

3. The authors should justify the virus strain that was used. Was there an association of this virus with casual or sexual contact?

The virus strain used was an isolate derived from the ZIKV outbreak in French Polynesia, which occurred just prior to the American outbreak. This virus shares a very recent common ancestor with strains currently circulating in the Americas (it shares 99% nucleotide identity with currently circulating viruses), and was chosen because it was known to infect macaques and was available to use immediately for this urgent study. To our knowledge, there are no strain-specific associations with whether ZIKV can be transmitted sexually or by casual contact. Sexual transmission was noted in the latest outbreak, but may have been overlooked in previous outbreaks because of their more limited scope and an epidemiological focus on mosquito transmission in tropical regions. The fact that, in the current epidemic, ZIKV was transmitted to people in areas where likely mosquito vectors were absent allowed sexual transmission to be identified more clearly. In addition, sexual transmission of ZIKV was first reported in association with an African strain (Foy et al. 2011, Emerging Infectious Diseases, PMID: 21529401), suggesting that sexual transmission may be a characteristic of ZIKV in general and not necessarily an adaptation of the currently circulating American strains. A justification for our choice of ZIKV strain has been added to the manuscript in the Methods section under "Virus" (lines 263-265).

4. The layout of the paper results is a little hard to follow. Might it be better to first directly compare the results from the two challenge models (subQ vs. oral)? And then talk about the saliva inoculations. As is,

the results from the SubQ come between the Oral and saliva challenge. At first glance the reader is inclined to think that an additional SubQ cohort was used for these studies but this is not the case.

We apologize for the confusion. To address this concern, the primary discussion of the subcutaneously infected animals was added to the first results paragraph just prior to comparing the viral loads of the animals infected at the tonsils vs. subcutaneously (lines 73-81). In addition, we clarified sections of text discussing the use of the subcutaneously infected animals as saliva donors to indicate that the animals are the same as those used to compare to tonsil-infected animals (lines 108-112).

5. The authors demonstrate nicely that the neutralizing titers at 28 dpi are similar between the SubQ and oral infection routes. However, the viremia seemed to lag a bit at early times post infection. Did the antibody responses also lag with similar kinetics? Analysis of antibody responses at additional early time points, by ELISA or neutralization assays, would help to determine whether the response times for Ab development were the same or offset by the viral lag period. In addition pre-infection serum for each animal should be used as a control.

We agree that this is an interesting question. Neutralizing antibody titers typically do not arise much earlier than 14 days post infection. After 10 days post-infection we only collected samples once/week, which is insufficient resolution to see a difference in kinetics of nAb titers, given that the lag in detectable viremia is only a day or two. Also, since viremia resolved in both groups at approximately the same time, the small lag in infection likely did not alter the kinetics of the neutralizing antibody response. We did use pre-infection serum as a control for the day 28 neutralizing assays, as the reviewer requested.

6. Were the first two groups of animals taken to necropsy to determine sites of virus infection? If so this data should be incorporated into the manuscript.

Animals were not euthanized for this study, so we do not have these data.

7. Often times in the text Figure references are not accurate. For example, lines 70, 79, 116—Fig. 2a should be Fig. 2b. There are similar discrepancies with other figure references within the text.

We apologize for these errors and have corrected the figure references.

8. While it is negative, the Ab titers data for the saliva challenged animals should be shown to prove to the reader that they did not become infected.

We agree that it is best to show the data, even though they are negative. These data are now presented in Supplementary Figure 1.

REVIEWERS' COMMENTS:

Reviewer #2 (Remarks to the Author):

The authors have addressed my concerns. They have provided new data that indicate that mixing saliva with ZIKV inhibits infectivity. They have added new sections in the results that help to clarify the two different models. They make a good point that doing viral titrations in the rhesus macaque model can be expensive and resource challenging. As this is unlikely to be a routine experimental challenge model at a minimum it gives a starting point for a viral dosage experiment.

Looking back through the manuscript, the only thing that may be important to add, if they have the data, would be in Supplemental Figure 2 a control set of mice that were infected with ZIKV that showed morbidity or mortality. These would simply be added as a control to demonstrate that the model is working as expected.

Response to Reviewer's Comments

Following our initial manuscript revision, reviewer 2 had one remaining comment (in italics), our response follows:

Looking back through the manuscript, the only thing that may be important to add, if they have the data, would be in Supplemental Figure 2 a control set of mice that were infected with ZIKV that showed morbidity or mortality. These would simply be added as a control to demonstrate that the model is working as expected.

We have added the control set of mice infected with ZIKV-FP to Supplemental Figure 2 as requested by the Reviewer.